# Integrative Analysis of Angiogenesis-Related Long Non-Coding RNA and Identification of a Six-DEARlncRNA Signature Associated with Prognosis and Therapeutic Response in Esophageal Squamous Cell Carcinoma

**DOI:** 10.3390/cancers14174195

**Published:** 2022-08-30

**Authors:** Shasha Cao, Xiaomin Wang, Xiaohui Liu, Junkuo Li, Lijuan Duan, Zhaowei Gao, Shumin Lun, Yanju Zhu, Haijun Yang, Hao Zhang, Fuyou Zhou

**Affiliations:** 1Henan Medical Key Laboratory, Precise Prevention and Treatment of Esophageal Cancer, Anyang Tumor Hospital, The Affiliated Anyang Tumor Hospital of Henan University of Science and Technology, Anyang 455000, China; 2Department of General Surgery, The First Affiliated Hospital of Jinan University, Jinan University, Guangzhou 510630, China

**Keywords:** ESCC, ARlncRNAs, risk model, prognosis

## Abstract

**Simple Summary:**

Esophageal squamous cell carcinoma (ESCC) is a familiar lethal malignance. Increasing evidence has disclosed that lncRNA is involved in tumorgenesis and progression in various tumor types. However, the current understanding of angiogenesis-related lncRNAs (ARlncRNAs) involved in ESCC remains evasive. We developed and validated a six-DEARlncRNA risk score system among the GSE53624 and GSE53622 set, aiming to identify the novel prognostic targets for ESCC. Our results showed that the six-DEARlncRNA could be used as an effective independent prognostic factor of ESCC. Furthermore, the six-DEARlncRNA biomarkers mainly participated in regulation of the skin and epidermis development, and these processes protected the body from environmental insults and may have been involved in the progression of ESCC. In conclusion, this study showed that the six-DEARlncRNA signature could be used as an independent prognostic factor and may be a valuable target for treatment options in ESCC.

**Abstract:**

Esophageal squamous cell carcinoma (ESCC) is a lethal gastrointestinal malignancy worldwide. We aimed to identify an angiogenesis-related lncRNAs (ARlncRNAs) signature that could predict the prognosis in ESCC. The GSE53624 and GSE53622 datasets were derived from the GEO database. The differently expressed ARlncRNAs (DEARlncRNAs) were retrieved by the weighted gene co-expression network analysis (WGCNA), differential expression analysis, and correlation analysis. Optimal lncRNA biomarkers were screened from the training set and the six-DEARlncRNA signature comprising AP000696.2, LINC01711, RP11-70C1.3, AP000487.5, AC011997.1, and RP11-225N10.1 could separate patients into high- and low-risk groups with markedly different survival. The validation of the reliability of the risk model was performed by the Kaplan-Meier test, ROC curves, and risk curves in the test set and validation set. Predictive independence analysis indicated that risk score is an independent prognostic biomarker for predicting the prognosis of ESCC patients. Subsequently, a ceRNA regulatory network and functional enrichment analysis were performed. The IC50 test revealed that patients in the high-risk group were resistant to Gefitinib and Lapatinib. Finally, the six DEARlncRNAs were detected by qRT-PCR. In conclusion, we demonstrated a novel ARlncRNA signature as an independent prognostic factor to distinguish the risk of ESCC patients and benefit the personalized clinical applications.

## 1. Introduction

Esophageal carcinoma is a common disease of the digestive tract worldwide [1]. Esophageal carcinoma is classified into two pathological types: esophageal adenocarcinoma and esophageal squamous cell carcinoma, which is widely distributed in Asia [2]. Despite the development of surgical and perioperative management in recent years, the 5-year outcome is still disappointedly low due principally to the difficulties in early diagnosis and tumor metastasis of ESCC [3,4]. Therefore, further study is required to identify more effective biomarkers with reliable clinical significance for ESCC, which will enhance the evaluation and efficiency of prognosis and therapy, and even improve our understanding of the underlying molecular mechanisms.

Angiogenesis is a complex physiological process involving numerous signaling pathways and active factors [5,6,7,8]. LncRNA play vital roles in angiogenesis by directly or indirectly regulating various molecules involving multiple key steps of angiogenesis [9]. Inhibition of angiogenesis can be beneficial in the prevention of the tumor growth and metastasis by preventing them from providing nutrients to tumor cells. As angiogenesis plays an essential role in cancer pathogenesis and treatment, novel therapeutic targets are required to develop against tumor angiogenesis and lncRNA can be extremely invaluable in this regard.

Both coding RNA and non-coding RNA (ncRNA) research demonstrated they are involved in the regulation of the various normal biological functions and the pathological processes. It was widely believed that ncRNAs were transcriptional noise initially [10], nevertheless, mounting evidence have indicated that they may regulate the various processes participating in the gene expression, multiple physiological and pathological processes [11,12,13,14]. Numerous studies indicated ncRNAs were frequently used to construct prognostic models of diseases in recent years owing to their stable expression in plasma, serum, and other body fluids [15,16]. These prognostic models are important for early diagnosis and therapy of diseases and have valuable important clinical significance. Considerable evidence have proved the important role of lncRNAs in both normal development and tumorigenesis [17,18]. It has been demonstrated that lncRNAs are involved in angiogenesis [19,20]. For example, overexpressing lncRNA CCAT2 significantly decreased the level of miRNA-424, increased the glioma cell growth, as well as promoted endothelial angiogenesis [21]. Lin et al. reported that lncRNA DANCR elevated ovarian tumor growth by promoting cancer cell angiogenesis [22]. The up-regulation of MYLK-AS1 elevated angiogenesis and ultimately enhanced the tumor progression in hepatocellular carcinoma [23]. Lnc-CCDST inhibits the angiogenesis and migration by degrading the DHX9 expression and destructing the binding of MDM2 to DHX9 in cervical cancer [24]. However, few studies have comprehensively explored angiogenesis-related lncRNA signature related prognostic model in ESCC.

In the present study, based on WGCNA and bioinformatics analysis, we identified DEARlncRNAs and constructed a risk score system of ESCC. Then, we conducted the prognostic value of the six DEARlncRNAs through KM survival, ROC, and risk curve analysis. Using univariate and multivariate regression analyses, the risk score was considered as an independent prognostic risk factor. Developing a nomogram consisting of the risk score and N stage or TNM stage can predict the ESCC patients’ survival probability extremely well. By calculating the Pearson correlation between the biomarkers and mRNA, we predicted the mRNAs targeted by biomarkers and constructed the ceRNA network. In addition, we stratified the high- and low-risk ESCC groups according to the risk score, identified the differentially expressed genes (DEGs), and ultimately conducted GO and KEGG analysis. Then, qPCR was used to detect the expression of the six DEARlncRNAs. In summary, this study shows that the six-DEARlncRNA signature could be an innovative prognostic tool in ESCC and perhaps offer alternative drug targets for clinical therapeutics.

## 2. Materials and Methods

### 2.1. Data Collection

The GSE53624 and GSE53622 datasets were obtained from the GEO database (https://www.ncbi.nlm.nih.gov/geo/ (accessed on 5 February 2022)). The GSE53624 dataset, including 119 ESCC patients and 119 normal samples, was used as a training set and test set. The GSE53622 dataset, including 60 ESCC patients and 60 normal samples, was used as a validation set [25]. The angiogenesis-related genes (ARGs) were gathered from the Genecards database (https://www.genecards.org/ (accessed on 5 February 2022)) and the Molecular Signatures Database (MsigDB) (http://software.broadinstitute.org/gsea/msigdb/ (accessed on 5 February 2022)). The search term was “Angiogenesis”. Then, 107 ARGs with “score > 5” and “Protein Coding” were screened in Genecards database, and 314 ARGs were screened in MsigDB database. Subsequently, the retrieval results were combined and duplicates were removed to obtain 363 ARGs. Among them, 63 ARGs were not discovered in the expression profile of the GSE53624 dataset. Therefore, a total of 300 ARGs were preserved for further study.

### 2.2. Identification of ESCC-Related lncRNAs Based on WGCNA Analysis

A WGCNA algorithm was performed to develop a co-expression network as well as to identify ESCC-related lncRNAs in the GSE53624 dataset [26]. In order to check the overall correlation of the samples, we first clustered the samples and eliminated the outlier samples to ensure the accuracy of the analysis. Next, we determined the soft threshold of the data, and verified k and P (k). In addition, the adjacency and similarity were calculated between lncRNAs, and we found the clustering tree. Meanwhile, the dynamic tree cutting algorithm was used to segment modules, and the minimum number of genes per module was 30. Finally, we evaluated the module-traits relationships between each module in ESCC patients and normal samples, and screened out the module lncRNA with the highest relations with ESCC.

### 2.3. Screening of Candidate DEARlncRNAs

The Limma package was used for screening the differently expressed lncRNAs (DElncRNAs) between ESCC patients and normal samples on 7247 lncRNAs in the GSE53624 dataset [27], with the following cutoff for adjustment: |log_2_FC| > 1 and *p* Value < 0.05. We used the ggplot2 [28] R package and the pheatmap R package to draw volcano maps and heat maps to demonstrate DElncRNAs expression, separately. Subsequently, we calculated the Pearson coefficient between ARGs and 7247 lncRNAs. According to the screening conditions with |Cor| > 0.7 and *p* < 0.01, we obtained the ARlncRNAs. Ultimately, candidate DEARlncRNAs were obtained by overlapping the ESCC-related lncRNAs, DElncRNAs, and ARlncRNAs using the Venn diagram.

### 2.4. Construction of a Risk Model

A total of 119 ESCC samples in GSE53624 dataset were randomly assigned 7:3, and 83 samples were used as the training set. The DEARlncRNAs were screened by univariate Cox regression analysis, and the univariate cutoff was set to a *p* value < 0.05. Then, the DEARlncRNAs were adopted to perform multivariate Cox regression, and the obtained DEARlncRNAs were used as prognostic biomarkers in this study [29]. The risk models were constructed using the predict.coxph function of survival package. Meanwhile, we calculated the risk score of the training set and the validation set. The prognostic formula was as follows: Riskscore = h0(t) × exp(β1 × 1 + β2 × 2 + … + βnXn). In this formula, β referred to the regression coefficient, for which the Hazard Rati (HR) value can be obtained after taking the inverse natural log exp(β), ho(t) was the baseline hazard function. The ESCC patients were separated into high- and low-risk groups based on the median risk score.

### 2.5. Validation of the Prognostic Risk Model

The survival time of the patients in the two groups was analyzed by K-M survival curves using the survminer package [30]. Then, ROC curves were adopted to calculate the AUC area to identify the specificity and sensitivity of the model, and the ROC curves were plotted using the survival ROC package with 1-, 3-, and 5-years as the survival time points [31]. Furthermore, the risk models were tested using the rest of 36 samples in the GSE53624 test set, and K-M survival curves, ROC curves, risk curves, and expression heat maps in high- and low-risk groups were also plotted. For verifying the applicability of the model better, we used the GSE53622 validation dataset to verify the risk models. Finally, we compared the expression levels of biomarkers in normal samples and ESCC samples in GSE53624 and GSE53622 datasets, respectively. The degree of difference was compared using wilcox.test method.

### 2.6. Independence Prognostic of the Risk Model

The correlation between risk score and clinical traits (age, sex, stage, T-stage, N-stage, alcohol, tobacco) in ESCC patients were evaluated by wilcox.test. To more closely understand the correlation between DEARlncRNAs and clinical traits, stratified survival analysis was used to investigate clinical traits for which the risk score had a difference [30]. The relationship between patients’ survival and clinical traits was then analyzed using the univariate and multivariate Cox regression analysis. We constructed nomogram based on risk scores and clinical data, and the calibration curves were used to observe deviations between predicted and actual survival rates.

### 2.7. Construction of a ceRNA Network

By calculating the Pearson correlation between the biomarkers and mRNA, we predicted the mRNAs targeted by biomarkers. The mRNAs with |Cor| > 0.7 and *p* < 0.01 were considered to be the mRNAs targeted by biomarkers. The miRWalk database (http://mirwalk.umm.uni-heidelberg.de/ (accessed on 5 February 2022)) was used to predict the related miRNA. We selected the miRNA predicted by miRtarBase and miRDB for subsequent analysis [32]. Meanwhile, we obtained the sequence of biomarkers lncRNA through lncipedia database (https://lncipedia.org/ (accessed on 5 February 2022)). Next, the miRDB database (http://mirdb.org/mirdb/index.html (accessed on 5 February 2022)) was used to predict biomarkers lncRNAs targeting of miRNAs. In this way, we obtained a binding network of lncRNA-miRNA-mRNA, which was called ceRNA regulation network.

### 2.8. Identification of DEGs and Functional and Pathway Enrichment Analyses

Firstly, the DEGs between high- and low-risk groups were screened using the limma package in the R platform [27], and genes with |log_2_FC| > 1, *p* value < 0.05 were regarded as DEGs. The DEGs were shown by volcano plots and heat map. The GO annotation and KEGG analysis were performed using the ClusterProfiler [33] software package in the R platform (*p* = 0.05, q = 0.2).

### 2.9. Drug Sensitivity Prediction

Chemotherapy was a common way to treat cancer, and drug sensitivity had a great impact on the therapeutic effect. To explore the sensitivity of drugs in patients with high- and low-risk groups, pRRophetic R package was applied to extrapolate IC50 by developing a ridge regression model with 10-fold cross validation [34,35]. Genomics of Drug Sensitivity in Cancer (GDSC) (https://www.cancerrxgene.org/ (accessed on 5 February 2022)) [36] was adopted to acquire 8 common chemotherapy drugs (Cisplatin, Paclitaxel, Gefitinib, Bosutinib, Erlotinib, Lapatinib, Bicalutamide, Vinorelbine) and their genetic profiles. Wilcoxon signed-rank test was applied to test the difference between groups, with *p* < 0.05 being a significant difference.

### 2.10. Validation of the Expression of the Six-DEARlncRNA by qRT-PCR

The carcinoma tissue and normal control of 10 patients were used for PCR to verify the expression of biomarkers. All participants signed informed consent prior to enrollment in the study. Study protocols were approved by the Ethics Committee of the Anyang Tumor Hospital, based on the ethical principles for medical research involving human subjects of the Helsinki Declaration. Total RNA was extracted using the Trizol reagent (cat.: 356281) provided by ambion company. Then, we used sweScript RT I First strand cDNA Synthesis All-in-OneTM First-Strand cDNA Synthesis Kit (cat.: G33330-50) from the Servicebio company for reverse transcription reaction. qRT-PCR was performed using the 2 × Universal Blue SYBR Green qPCR Master Mix kit (cat.: G3326-05) provided by Servicebio. Primer sequences were shown in Appendix A. GAPDH was used as an internal reference gene. The qRT-PCR reactions were as follows: 95 °C pre denaturation for 1 min, and then 40 cycles of 95 °C denaturation for 20 s, 55 °C annealing for 20 s, and 72 °C extension for 30 s. Additionally, the relative mRNA expression was normalized and calculated according to the 2^−^^△△Ct^ method.

## 3. Results

### 3.1. Identification of ESCC-Related lncRNA Based on WGCNA Analysis

The WGCNA was used to analyze the ESCC-related lncRNA based on the GSE53624 dataset. Firstly, we found that the clustering of samples were clear and there were no outlier samples needed to be deleted (Figure 1A). When the ordinate scale-free R^2^ approached the threshold value of 0.85 (red line), the network was close to scale-free distribution and mean connectivity was close to 0 (Figure 1B). There were 33 modules obtained, and the cluster dendrogram and clustering plot of module eigengenes were displayed in Figure 1C. Next, the correlation between modules and clinical characteristics was analyzed. The results showed that MElightcyan1 module had the most significant correlation with the ESCC patients (*p* = 3 × 10^−104^) (Figure 1D) and this module contained 4448 lncRNAs (Figure 1E). We used MElightcyan1 as a key module for subsequent analysis.

### 3.2. Screening of Candidate DEARlncRNAs

We acquired the DElncRNAs expression pattern of the GSE53624 dataset by the Limma R package. A total of 777 were obtained between ESCC patients and normal samples, of which 341 DElncRNAs were upregulated and 436 DElncRNAs were downregulated (Figure 2A). Then, the *p* value was ranked in ascending order, and the top 100 were selected to draw a heatmap of DElncRNAs (Figure 2B). The Pearson correlation was evaluated between 300 ARGs and 7247 lncRNAs in the GSE53624 dataset and a total of 1337 ARlncRNAs were obtained. Subsequently, 183 candidate DEARlncRNAs were obtained by overlapping the 4448 ESCC-related lncRNAs, 777 DElncRNAs, and 1337 ARlncRNAs (Figure 2C).

### 3.3. Construction of a Risk Model

To investigate the effect of key ARlncRNAs on ESCC prognosis, 183 DElncRNAs were input in univariate Cox regression on 83 ESCC samples from the GSE53624 dataset to identify robust markers. In total, 10 DEARlncRNAs were retrieved and the forest map was shown in Figure 3A. Subsequently, the prognostic biomarkers were established using multivariate Cox regression analysis and 6 DEARlncRNAs (AP000696.2, LINC01711, RP11-70C1.3, AP000487.5, AC011997.1, RP11-225N10.1) were obtained. Furthermore, we found that AP000696.2, LINC01711, AP000487.5, and AC011997.1 were risk factors with HR > 1, while RP11-70C1.3 and RP11-225N10.1 were favorable factors with HR < 1 (Figure 3B).

### 3.4. Validation of the Prognostic Risk Model

To evaluate the prognostic prediction of risk genes in ESCC patients, we calculated the risk score of each patient. Patients in the training set were separated into a high-risk group (n = 42 patients) and a low-risk group (n = 41 patients) based on the median risk score of 0.860189 (see the Materials and Methods section for details). The Kaplan-Meier survival analysis showed that the prognosis of the high- and low-risk groups was significantly different, and the OS of the high risk group was very short (Figure 4A). Then, the ROC curve was plotted to predict the 1-, 3-, and 5-year survival and the results were shown in Figure 4B. The results showed that the 1-, 3-, and 5-year AUCs were 0.768, 0.825, and 0.822, respectively. Therefore, the risk model could effectively serve as a prognostic model. Figure 4C presented the risk score distribution. The expression of six risk genes between high- and low-risk groups in the GSE53624 training set was shown in Figure 4D. Besides, the six-DEARlncRNA risk model was used to analyze the test set and the validation set. Using the remaining 36 samples in the GSE53624 and GSE53622 as test set and validation set, the survival curves (Appendix A), the ROC curves (Appendix A), and the risk score panels (Appendix A) between the high-risk groups and the low-risk groups were significantly different. These results indicated that the risk model can be used as an effective and reliable predictor for ESCC patient’s prognosis.

### 3.5. Independence Prognostic of the Risk Model

Univariate and multivariate Cox regression analyses were used to evaluate the independent predictive values of the six-DEARlncRNA signature in ESCC patients. Despite univariate cox analysis of the GSE53624 and GSE53622 dataset suggested risk score, the pathological stage and N stage were related to the prognosis of ESCC patients (Figure 5A and Appendix A), the multivariable cox regression analyses showed only the risk score was an independent prognostic indicator (Figure 5B and Appendix A). We performed a stratified survival analysis of high- and low-risk groups across clinical traits including different TNM stages and age. It showed that the risk score based on the six-DEARlncRNA signature in GSE53624 could effectively predict OS in all subgroups from different clinical traits, including age ≤ 59 (*p* < 0.0001), age > 59 (*p* = 0.0037), stages I–II of TNM (*p* < 0.0001), stage III (*p* = 0.00059), N0 and N1 (*p* < 0.0001), N2 and N3 (*p* = 0.014), T1 and T2 (*p* = 0.005), T3 and T4 (*p* < 0.0001) (Figure 5C). The survival analysis of all subgroups showed differences in outcome between the low-risk and high-risk group based on the risk score in GSE53622 (Appendix A). Meanwhile, the nomogram were constructed in ESCC patients from GSE53624 and GSE53622 dataset (Figure 6A and Appendix A). It had good discrimination performance with the C-index of 0.6937 in the GSE53624 and 0.6599 in GSE53622. The calibration curve analysis showed good agreement with the observed survival probability in GSE53624 and GSE53622 (Figure 6B and Appendix A).

### 3.6. Construction of a ceRNA Network

Firstly, we identified 927 mRNAs targeted by the six DEARlncRNAs through Pearson correlation analysis. Next, the miRNAs were predicted based on 927 mRNAs that have been identified. The miRDB database was used to predict miRNAs targeted by biomarker lncRNAs. Ultimately, the ceRNA regulatory network was constructed, which contained 218 nodes and 300 sides (Figure 7). The network contained 6 biomarker lncRNAs, 107 miRNAs, and 105 mRNAs.

### 3.7. Identification of DEGs and Functional Enrichment Analyses

There were 98 DEGs between high- and low-risk groups, of which 17 DEGs were upregulated and 81 DEGs were downregulated (Figure 8A). The DEGs heat map was shown in Figure 8B. KEGG pathway and GO function analyses were conducted. In total, 54 enriched GO terms were obtained and the top 10 were shown in Figure 8C, including skin development, epidermis development, primary alcohol metabolic process, and retinol metabolic process. Meanwhile, the DEGs were significantly enriched in arachidonic acid metabolic pathway (Appendix A).

### 3.8. Drugs Sensitivity Prediction

IC50 was used to assess chemotherapeutic drug (cisplatin, paclitaxel, gefitinib, bosutinib, erlotinib, lapatinib, bicalutamide, and vinorelbine) sensitivity. The results showed that the IC50 of Gefitinib and Lapatinib were significantly different in the high- and low-risk groups (Figure 9). It can be indicated that Gefitinib and Lapatinib were more sensitive in the low-risk group. Therefore, Gefitinib and Lapatinib were not recommended for high-risk patients to chemotherapy.

### 3.9. Validation of the Gene Expression by qRT-PCR

We investigated the expression of the six DEARlncRNAs from the GSE53624 and GSE53622 dataset by wilcox.test. Relative to its expression level in normal tissue, LINC01711, AP000487.5, and AC011997.1 were remarkably higher, while AP000696.2, RP11-70C1.3, and RP11-225N10.1 were significantly down-regulated in GSE53624 ESCC tissues (Figure 10A). Likewise, the changing trend of the expression level of the six DEARlncRNAs in GSE53622 dataset was concordant with the results in GSE53624 dataset (Figure 10B). In order to further verify the expression of the biomarkers, we used qRT-PCR to compare gene expression levels in carcinoma tissue and normal control of 10 patients. The expression level of all six DEARlncRNAs detected by qRT-PCR was completely consistent with the results of data mining (Figure 10C).

## 4. Discussion

For most types of cancers, including ESCC, clinical and pathologic features are the traditional factors to give insight into potential strategies for treatment and still act as a prognostic predictor. However, considering the high heterogeneity of ESCC, investigation of novel and reliable molecular biomarkers and models for ESCC early diagnosis and prognosis is necessary. Angiogenesis is related to the development of tumor growth and metastasis and thus inhibiting angiogenesis has been a novel ESCC anticancer agents [37]. In esophageal cancer, many studies on lncRNA expression profile have been reported [38,39], but no angiogenesis-related signature has been applied in this disease at present. Therefore, elucidating a prognostic angiogenesis-related lncRNA signature is imperative and meaningful for ESCC patients.

We first employed WGCNA to generate an ESCC-related lncRNA in GSE53624 database and identified a total of 183 prognosis-related DEARlncRNAs by overlapping DElncRNAs, ARlncRNAs and WGCNA modulelncRNAs. Later, we developed a six-DEARlncRNA model including AP000696.2, LINC01711, RP11-70C1.3, AP000487.5, AC011997.1 and RP11-225N10.1 and this model can divide patients into high and low risk groups in GSE53624 training set. The performance of this six-DEARlncRNA model was further analyzed in the GSE53624 test set and GSE53622 validation set through KM, ROC survival curve and risk curve analysis. Moreover, multivariable analysis confirmed that the risk score of this prognosis signature was considered as an independent prognostic risk factor. Next, we further constructed a prognostic nomogram combined signature with N stage or TNM stage for disease assessment in ESCC patients. Compared with the actual survival rate, the calibration curve showed moderate consistency. Besides, IC50 detection in high- and low-risk patients showed that there were significant differences between Gefitinib and Lapatinib group, showing that the high-risk group were resistant to Gefitinib and Lapatinib. The lncRNA-mRNA-miRNA network analysis was an effective way to analyze the molecular regulation mechanism and explore the function. A prognosis-related ceRNA regulatory network was established in ESCC consisting of 6 lncRNAs, 107 miRNAs, and 105 mRNAs. Correlation analysis showed that the risk score could predict the effectiveness of chemotherapy agents. In summary, our study indicated that the six-DEARlncRNA had predictive value for diagnosis and treatment in ESCC.

Among the six DEARlncRNAs that composed the prognostic model, AP000696.2, LINC01711, and AC011997.1 had been reported as potential biomarkers in different cancers. Evidence suggested that a two-lncRNA signature, including AP000696.2 and ADAMTS9-AS1, had better predictive performance than traditional tumor markers [40]. The functional enrichment analysis of the two-lncRNA signature revealed that they may be involved in the ectoderm and epidermis development, coinciding with our results. Development and differentiation of the ectoderm into the epidermis was a key step in normal development and cancer progression, and the two novel lncRNAs might be involved in the ESCC tumorigenesis and progression [41]. Xu et al. revealed that LINC01711 was upregulated in esophageal carcinoma and associated with malignant phenotypes and poor prognosis [42]. Existing literature had shown that LINC01711 was included in the prognostic model and may play essential roles in malignant progression of ESCC, triple negative breast cancer and fibrosis [43,44,45]. Shi et al. reported that AC011997.1 was considered as a prognostic risk factor in ESCC [30].

In contrast to other prognostic models, we mainly focused on the lncRNA associated with angiogenesis and established a six-DEARlncRNA signature prognosis model in ESCC. Previous studies have developed several prognosis signature models based on lncRNA expression in ESCC. Thus, our prognosis model had dual functions since it could indicate the patient’s prognosis and contribute to select a more appropriate form of antiangiogenesis treatment for patients. However, lncRNAs were involved in the metabolism, immune response, and multiple physiological processes, not only in the angiogenesis. Comparing global investigation of lncRNAs related to ESCC prognosis, our study screened angiogenesis-related lncRNA, which limited new discoveries associated with other biological characteristics and may not provide more broadly potential clinical solutions.

Nonetheless, there are still some limitations in this study. Firstly, although we have verified the prognostic values and predictive accuracy of the six-DEARlncRNA risk model in 179 established ESCC patients, and qRT-PCR was used to verify the expression of six-DEARlncRNA, external validation datasets are still not included in the model. Employing more dataset and bioinformatics strategies is the way to verify and advance the model in the future. Secondly, the fundamental experiments are needed to support the proposed DEARlncRNA signature is involved in angiogenesis. Furthermore, drug sensitivity prediction results is based on public databases and additional in vitro and in vivo experiments are needed to validate the prediction performance. Finally, the specific functions and potential molecular mechanisms of the prognostic model need to be further explored.

## 5. Conclusions

In conclusion, our study identified that the six-DEARlncRNA risk model, including AP000696.2, LINC01711, RP11-70C1.3, AP000487.5, AC011997.1, and RP11-225N10.1, is a faithful tool in predicting the prognosis of ESCC patients. In addition, this model can distinguish and indicate the sensitivity of chemotherapy drugs (Gefitinib and Lapatinib) in ESCC, thus preventing patients from suffering unnecessary drug side effects. Furthermore, the established nomogram with good consistency can help to predict ESCC patients’ survival and outcome accurately, thus providing clinical strategies for clinicians.

## Figures and Tables

**Figure 1 cancers-14-04195-f001:**
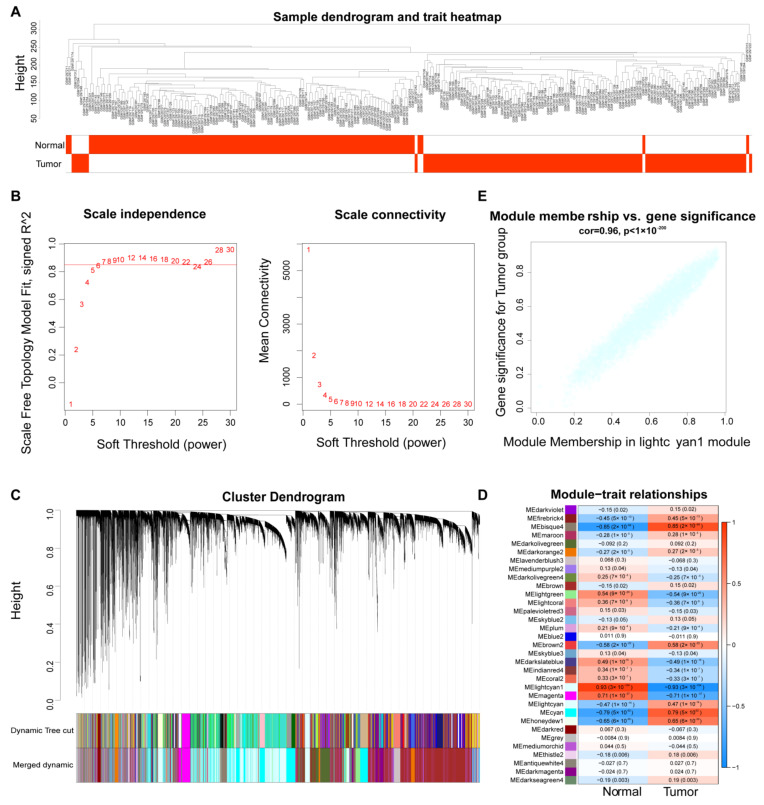
WGCNA analysis identified ESCC-related lncRNAs; (**A**) Clustering dendrogram of samples with trait heatmap; (**B**) Graphs of scale-free topology model fit index and mean connectivity versus soft threshold. Seven was chosen as the most fit soft-power value; (**C**) Clustering dendrogram of all differentially expressed lncRNAs; (**D**) Analysis of the module-trait relationships between ESCC and normal samples. The MElightcyan1 module was the most relevant to tumor; (**E**) Scatter plot of module eigengenes in the MElightcyan1 modules.

**Figure 2 cancers-14-04195-f002:**
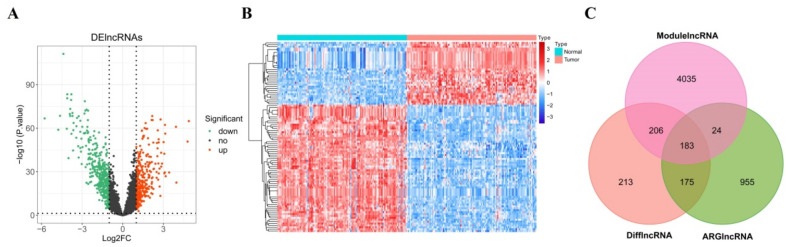
Screening the prognostic DEARlncRNAs in ESCC patients; (**A**) Volcano plot visualized the differential expression genes between ESCC and normal samples. The upregulated gene is displayed in red dot and the downregulated gene is in green; (**B**) The heat map of the top 100 DElncRNAs. The row and column represent genes and samples, respectively; (**C**) Venn plots reveal the overlapping 183 DEARlncRNAs of the DElncRNAs, ARlncRNAs, and WGCNA modulelncRNAs.

**Figure 3 cancers-14-04195-f003:**
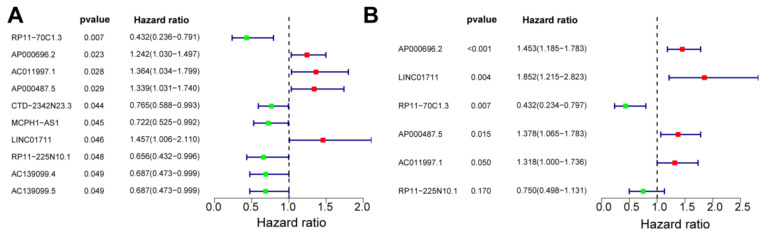
Forest map of univariate Cox regression analysis (**A**) and multivariate Cox regression analysis (**B**) for DEARlncRNAs.

**Figure 4 cancers-14-04195-f004:**
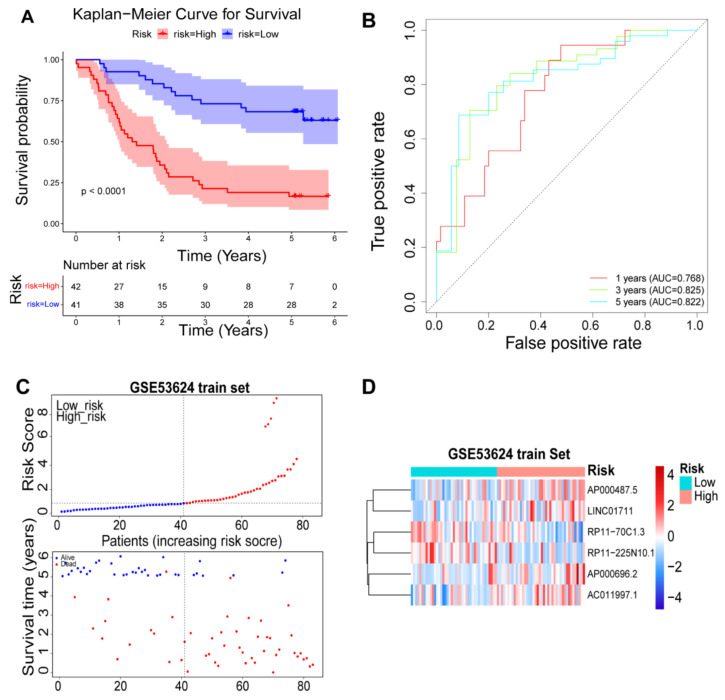
Prognostic values and predictive accuracy of the six-DEARlncRNA. K-M survival curve (**A**); time-dependent ROC curve (**B**); risk score distribution ((**C**), **above**); survival overview ((**C**), **below**); and heatmap (**D**) for patients in the GSE53624 training set assigned to high- and low-risk groups based on the risk score.

**Figure 5 cancers-14-04195-f005:**
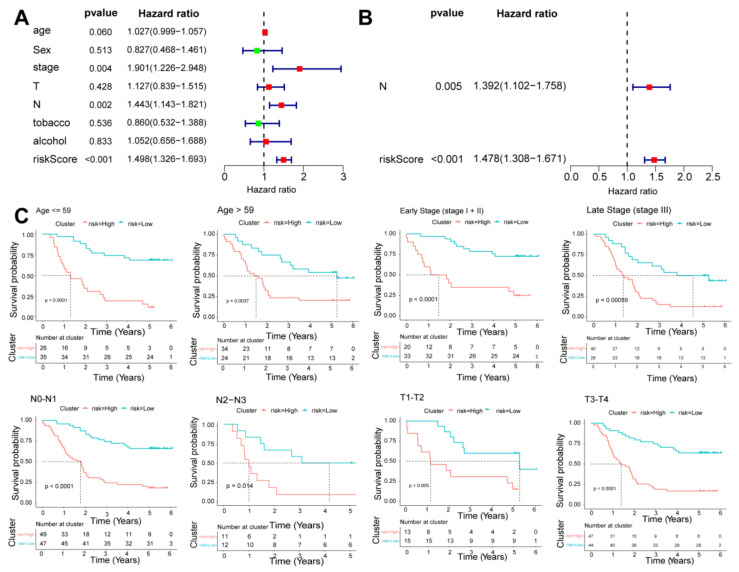
A stratified K-M survival analysis across different clinical features in GSE53624. Univariate (**A**) and multivariate (**B**) Cox regression analyses of the association between clinicopathological factors and overall survival (OS) of patients in GSE53624 dataset. (**C**) K-M survival curves for OS in multiple subgroups based on the six-DEARlncRNA risk score.

**Figure 6 cancers-14-04195-f006:**
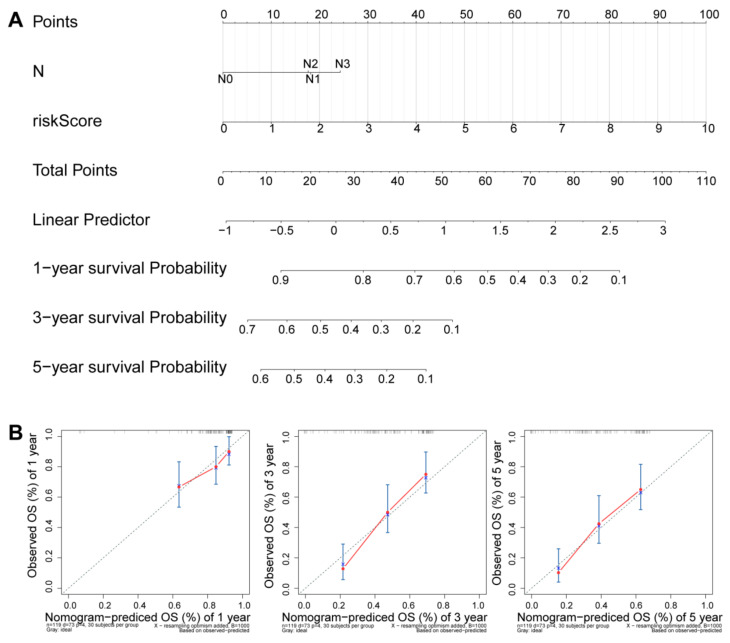
Establishment of a nomogram for survival prediction in GSE53624; (**A**) Nomogram model consisting of N stage and risk score factors for 1-, 3-, and 5-year OS prediction; (**B**) Calibration curves of the nomogram model showing the predicted survival probabilities and the actual observed proportions.

**Figure 7 cancers-14-04195-f007:**
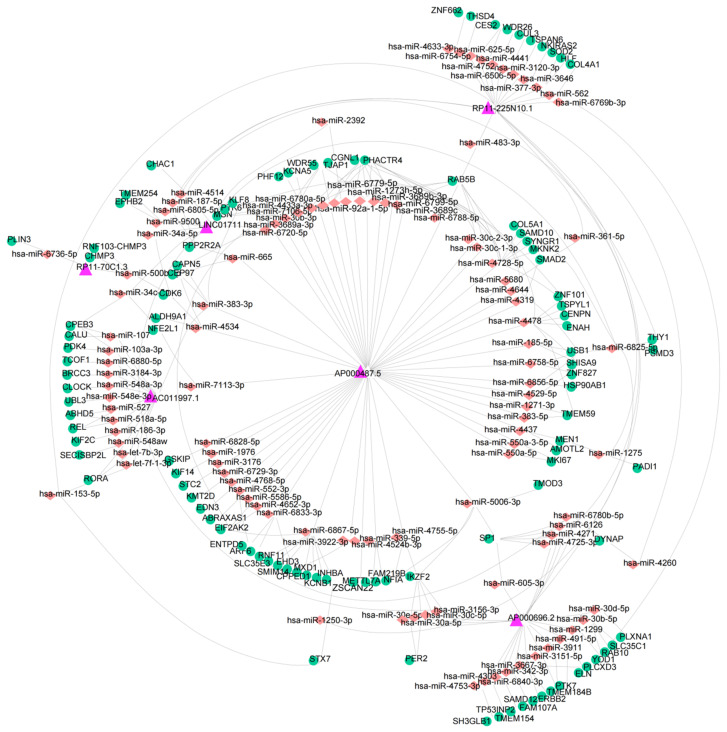
The ceRNA network was constructed of the six prognosis-related DEARlncRNAs (purple) and their target miRNAs (orangered) and mRNAs (green).

**Figure 8 cancers-14-04195-f008:**
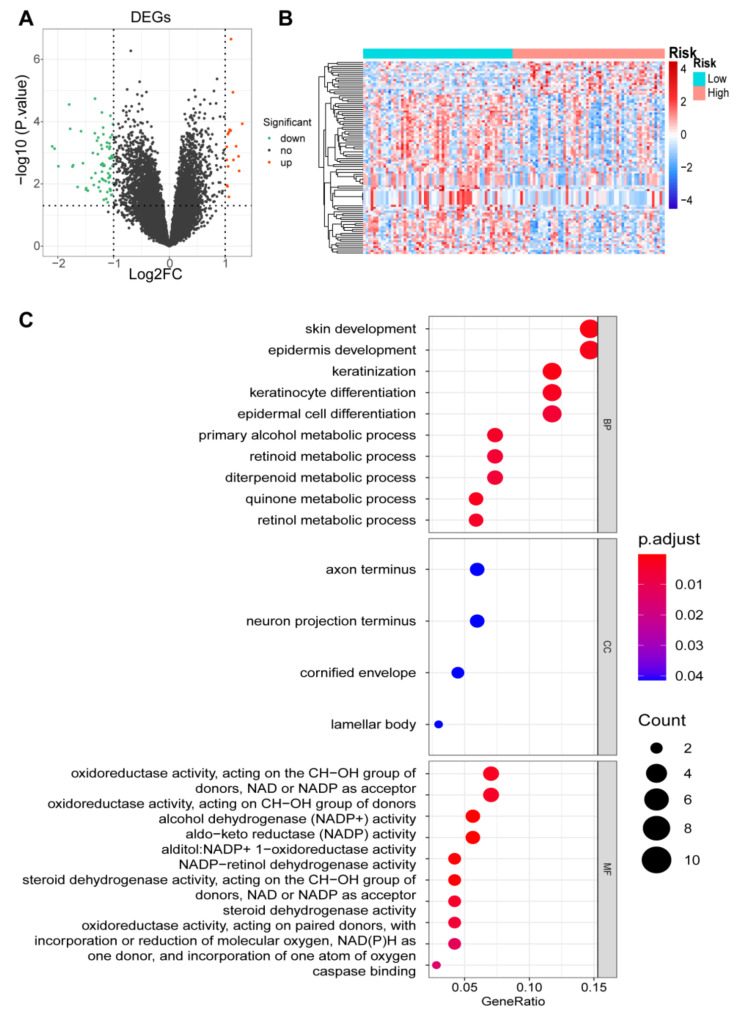
DEGs functional enrichment analysis; (**A**) The volcano plot of the upregulated and downregulated DEGs; (**B**) Heatmap for the hierarchical cluster analysis of high- and low-risk sample; (**C**) Top 10 enriched GO terms for DEmRNAs.

**Figure 9 cancers-14-04195-f009:**
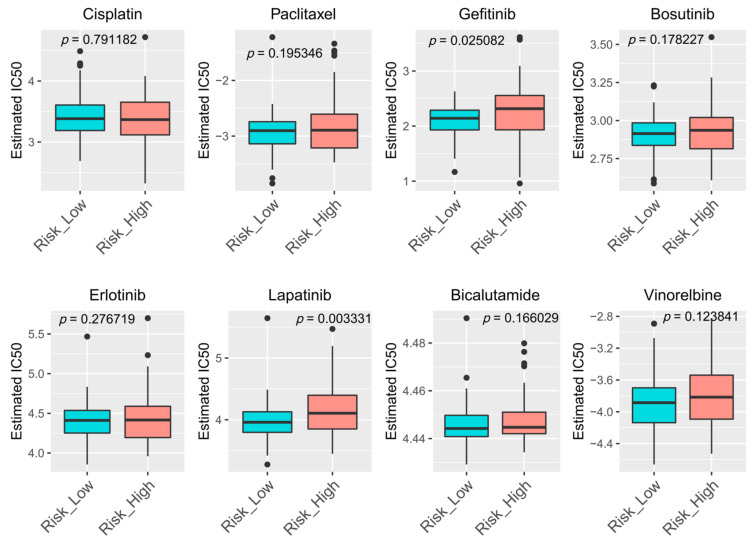
IC50 value of eight common chemotherapeutic drugs showing there were significant differences between Gefitinib and Lapatinib group. Blue box indicated low risk group, orange box indicated high risk group, and black dots indicated discrete samples.

**Figure 10 cancers-14-04195-f010:**
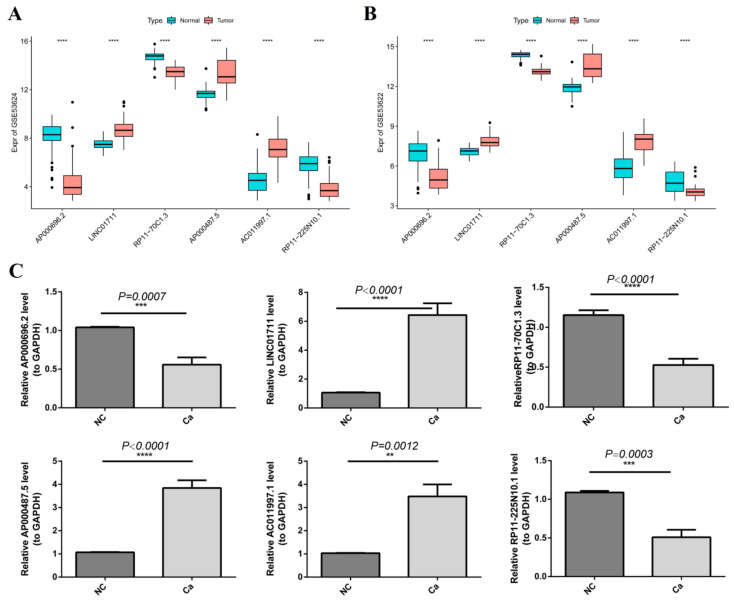
The expression of the six-DEARlncRNA was verifed by qRT-PCR. (**A**,**B**) The mRNA expression levels of the six-DEARlncRNA signature in 179 cancer samples and corresponding adjacent samples from the GSE53624 ((**A**), n = 119) and GSE53622 dataset ((**B**), n = 60). Black dots indicated discrete samples. (**C**) Expression of six-DEARlncRNA in 10 pairs of ESCC and adjacent normal tissues by qRT-PCR. ** *p* < 0.01; *** *p* < 0.001; **** *p* < 0.0001.

## Data Availability

The datasets generated and/or analyzed during the current study are available from the corresponding author on reasonable request.

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
