# Peer review of "Integrative Analysis of Angiogenesis-Related Long Non-Coding RNA and Identification of a Six-DEARlncRNA Signature Associated with Prognosis and Therapeutic Response in Esophageal Squamous Cell Carcinoma"

_cancers, 2022, doi:10.3390/cancers14174195_

Round 1

Reviewer 1 Report

Authors performed screening of candidate DEARlncRNAs. Volcano plot, heat map and Venn plots were analyzed. DEARlncRNAs (AP000696.2, LINC01711, RP11-70C1.3, AP000487.5, AC011997.1, RP11-225N10.1) were selected to construct a risk mode. Consequently, authors validated prognostic risk model by calculating ROC curve. Finally, authors validated by univariate and multivariate Cox regression analyses of the association between clinicopathological factors and overall survival (OS) of patients in GSE53624 dataset. The structure of validations is acceptable. lncRNAs are involved in some important cancer function such as angiogenesis. Thus, newly discovered DEARlncRNA signatures might be an innovative prognostic tool in ESCC. However, there some concerns as follows.

Major comments

・The resolution of figures is not enough. Readers were difficult to read the characters. Please use more fine resolution especially in Figure 3, 4 and 7.

・Drug sensitivity prediction study is difficult to understand. How to analyze and the reliability are unclear. As authors described “To explore the sensitivity of drugs in patients with high- and low-risk groups, IC50 value of 8 common chemotherapy drugs (Cisplatin, Paclitaxel, Gefitinib, Bosutinib, Erlotinib, Lapatinib, Bicalutamide, Vinorelbine) were analysed using the pRRophetic R package.” in the material method section. I think that pRRophetic is an R package for prediction of clinical chemotherapeutic response from tumor gene expression levels. However, I think that many readers who is interested in esophageal cancer cannot catch up it. Authors should explain more carefully about this section. Moreover, it is unclear how the drugs are really effective because it is merely a prediction model. Authors should describe the limitation about drug sensitivity prediction.

Minor comments

・Font size was not equal. Please adjust the size.

Reviewer 2 Report

Cao, Wang et al reanalysed two long non-coding (lnc)RNAs RNA-seq datasets, GSE53622 and GSE53624 (part of the same SuperSeries GSE536250) in normal or oesophageal squamous cell carcinoma (ESCC). The authors used these two datasets to find six lncRNAs that are differentially expressed in tumour versus control and whose expression is correlated with angiogenesis protein-coding genes (called DEARlncRNAs). This six-DEARlncRNAs signature can separate patients into low- and high-risk with different survival time and sensitivity to chemotherapy drugs. The authors also confirmed by qRT-PCR the differential expression of the six DEARlncRNAs in tumour compared to control.    

Globally, I think the paper is interesting and I do not have many comments.

Comments:

1.     I think the authors should also add as a limitation that the six DEARlncRNAs have not been tested experimentally as being associated with angiogenesis. Currently, the authors found a correlation between these lncRNAs and protein-coding genes associated in angiogenesis. Knockdown/knockout experiments will be needed to show a role of these six DEARlncRNAs in angiogenesis.

2.     Another recent paper from other authors have reanalysed the same dataset (GSE53625, which includes GSE53622 and GSE53624) and found an eight lncRNAs signature that predicts prognosis of ESCC (Zhang J et al, Identification and validation of an eight-lncRNA signature that predicts prognosis in patients with esophageal squamous cell carcinoma. Cellular & Molecular Biology Letters, 2022). There is no overlap between the eight lncRNAs that have been found in this paper and the six DEARlncRNA signature found in the current manuscript, likely due on the focus in this manuscript on lncRNAs whose expression correlate with angiogenesis protein-coding genes. The authors should discuss the advantages and inconveniences of using a more specific biological function (i.e. angiogenesis) compared to a global investigation of lncRNAs that could act as an ESCC-signature.

Round 2

Reviewer 1 Report

My concern was disappeared. Thank you for accurate revising.